



# Heavy-tailed flood peak distributions: What is the effect of the spatial variability of rainfall and runoff generation?

Elena Macdonald[1], Bruno Merz[1,2], Viet Dung Nguyen[1], and Sergiy Vorogushyn[1]

[1]GFZ German Research Centre for Geosciences, Potsdam, Germany.
[2]Institute for Environmental Sciences and Geography, University of Potsdam, Germany.

**Correspondence:** Elena Macdonald (elena.macdonald@gfz-potsdam.de)

**Abstract.** The statistical distributions of observed flood peaks often show heavy tail behaviour, meaning that extreme floods are more likely to occur than for distributions with exponentially receding tail. Falsely assuming light-tailed behaviour can lead to an underestimation of extreme floods. The robust estimation of the tail is often hindered due to the limited length of time series. Therefore, a better understanding of the processes controlling the tail behaviour is required. Here, we analyse how the spatial

variability of rainfall and runoff generation affect the flood peak tail behaviour in catchments of various size. This is done using a model chain consisting of a stochastic weather generator, a conceptual rainfall-runoff model and a river routing routine. For a large synthetic catchment, long time series of daily rainfall with varying tail behaviour and varying degree of spatial variability are generated and used as input for the rainfall-runoff model. In this model, the spatial variability and mean depth of a subsurface storage capacity are varied, affecting how locally or widely saturation excess runoff is triggered. Tail behaviour is

characterized with the shape parameter of the Generalized Extreme Value (GEV) distribution. Our analysis shows that smaller catchments tend to have heavier tails than large catchments. Especially for large catchments, the GEV shape parameter of flood peak distributions was found to decrease with increasing spatial rainfall variability. This is most likely linked to attenuating effects in large catchments. No clear effect of the spatial variability of the runoff generation on the tail behaviour was found.

## 1 Introduction

Extreme floods often come as a surprise and many examples of surprising floods can be found in the literature (Merz et al., 2015). This is partly because of the low occurrence probability of extreme floods. Human intuition tends to expect light-tailed behaviour (Taleb, 2007) and this would mean that extremes are very unlikely. However, when a distribution is heavy-tailed, rather than light-tailed, the occurrence probability of extreme flood events is much higher. The upper tail of a distribution is called heavy when it decreases slower than exponentially, making the occurrence of extremes more likely (El Adlouni et al.,

2008; Papalexiou and Koutsoyiannis, 2013). One example is the surprising and devastating flood that happened in the Ahr valley in west Germany in summer 2021. The flood peak distribution which was used to derive flood hazard maps before the 2021 event was almost light-tailed, suggesting an extremely low occurrence probability (return period > 1 million years) for floods of the magnitude of the 2021 event. However, considering historical floods in the same area results in a flood peak





distribution that is extremely heavy-tailed (Vorogushyn et al., 2022). In fact, many streamflow and precipitation time series
exhibit heavy tail behaviour (Bernardara et al., 2008; Farquharson et al., 1992; Smith et al., 2018; Villarini et al., 2011).

To quantify the tail-heaviness of a distribution, different indices exist (Wietzke et al., 2020). One that is frequently used in
hydro-meteorological studies is the shape parameter of the generalized extreme value (GEV) distribution. The GEV distribu-
tion is the asymptotic distribution of independent block maxima (Fisher and Tippett, 1928) and widely accepted as suitable
distribution for annual maximum series. When the shape parameter of a GEV distribution is larger than zero, the distribution
is considered to be heavy-tailed (El Adlouni et al., 2008). The estimation of the upper tail behaviour is however associated
with high uncertainties as it is sensitive to the largest few events (Merz and Blöschl, 2009). This is especially true for the short
observational time series that are typically available. Possible ways to achieve more robust estimations of the tail behaviour
can be regionalization approaches, the inclusion of historical floods, the simulation of long time series, and improving the
understanding of processes that control the tail behaviour (e.g. Merz and Blöschl, 2005; Vorogushyn et al., 2022; Macdonald
et al., 2024).

Several studies have addressed the potential controls of heavy tail behaviour and related characteristics of flood peak distri-
butions (Merz et al., 2022). These range from data-based approaches (e.g. Macdonald et al., 2022; Thorarinsdottir et al., 2018;
Villarini and Smith, 2010) to model-based approaches (e.g. Struthers and Sivapalan, 2007; Rogger et al., 2013; Macdonald
et al., 2024) and review studies (e.g. Merz et al., 2022). While some of the studies looked specifically at indicators of the tail
behaviour like the GEV shape parameter (e.g. Macdonald et al., 2024; Villarini and Smith, 2010), others considered the entire
flood frequency curve (e.g. Struthers and Sivapalan, 2007; Rogger et al., 2013) or flood skewness (McCuen and Smith, 2008;
Merz and Blöschl, 2009).

In their review, Merz et al. (2022) formulated 9 hypotheses on the influence of atmospheric, catchment and river network
factors on flood peak tail behaviour. Previous studies have linked characteristics of rainfall (Gaume, 2006), runoff generation
(Macdonald et al., 2022) and catchment characteristics such as size (Villarini and Smith, 2010) or aridity (Farquharson et al.,
1992) to the tail behaviour of flood peak distributions. Especially rainfall characteristics and runoff generation processes have
been found relevant and their influence on the tail behaviour is closely interlinked (McCuen and Smith, 2008; Macdonald
et al., 2024). Macdonald et al. (2024) concluded that for both aspects, i.e. rainfall and runoff generation, the spatial structure
can strongly affect flood peak generation and should therefore also be considered in the context of flood peak tail behaviour.

For small homogeneous catchments, Macdonald et al. (2024) found that the rainfall tail dominates the flood peak tail beyond
a certain return period. Similarly, Gaume (2006) stated that the statistical properties of rainfall asymptotically control the shape
of flood peak distributions. However, it is not clear from those studies whether this also holds for spatially variable rainfall.
Using simulation-based approaches, it has been found that the influence of rainfall spatial variability on floods increases with
return period (Peleg et al., 2017; Zhu et al., 2018). This indicates that it might also affect the tail behaviour. Haberlandt and
Radtke (2011) found differences in derived flood probabilities when different degrees of spatial rainfall variability (uniform
vs. random) were used to force a hydrological model. While they do not state a clear impact of rainfall spatial variability
on flood peak tail behaviour, their Fig. 4b suggests a heavier tail for spatially uniform than for variable rainfall. In contrast,
Wang et al. (2023) concluded, based on simulations for 5 catchments in Germany, that increasing spatial variability of rainfall



results in heavier tails of flood peak distributions beyond a certain degree of variability. They argued that spatially variable

rainfall creates more opportunities for partial saturation excess because the soil moisture becomes more heterogeneous than with spatially uniform rainfall. So on the one hand, spatially variable rainfall could lead to more variability of resulting flood flows and a higher chance of floods that are significantly larger than the bulk of the events, which would make a heavy-tailed distribution likely. On the other hand, spatially variable rainfall could also lead to an attenuation of flood peaks and thus reduce the differences between small and large events, which would make a light-tailed distribution likely.

Runoff generation can affect the tail behaviour of flood peak distributions especially through threshold processes (Macdonald et al., 2024). Threshold processes in the runoff generation have also been found to result in step changes in flood frequency curves (FFCs) (Rogger et al., 2012; Struthers and Sivapalan, 2007). In a data-based study of 480 catchments in Germany and Austria, Macdonald et al. (2022) found that heavy-tailed flood peak distributions emerge especially when there are distinct differences in the catchment response between small and large flood events. Such distinct differences and step changes have

been found to decrease with a more variable spatial distribution of storage capacities in a catchment (Rogger et al., 2013). Similarly, Struthers and Sivapalan (2007) found that spatially variable soil depth "simplifies" FFCs by allowing a smooth instead of an abrupt transition from unsaturated to saturated. They stated that homogeneous conditions lead to a binary saturation behaviour, while heterogeneous conditions lead to a steadily increasing partial saturation. Basso et al. (2015) linked heavy tails of streamflow distributions to highly nonlinear storage-discharge relations, and in a follow-up study suggested that the

spatial heterogeneity of hydraulic properties between hillslopes is one factor leading to such nonlinear relationships (Basso et al., 2016). Based on these studies, Merz et al. (2022) concluded in their review that while the importance of the catchment response for heavy-tail behaviour is clear, "there are contrasting answers to the question whether spatial variability in runoff generation enhances or dampens the tail heaviness" of flood peak distributions.

Catchment size has been found to influence flood peak tail behaviour, and it also interacts with the spatial variability of

rainfall and runoff generation. In data-based studies, tail heaviness has been found to decrease with increasing catchment size in Austria (Merz and Blöschl, 2009), Germany and Austria (Macdonald et al., 2022), and the eastern United States (Villarini and Smith, 2010). In contrast, studies in the Appalachian region (Morrison and Smith, 2002) and the entire United States (Smith et al., 2018) did not find an effect of the catchment size on flood peak tail behaviour. Catchment size could influence the tail behaviour in different ways. First, the dominant factors in flood generation can shift with catchment size: e.g., convective

rainfall bursts are more relevant for small catchments, while the effects of flood routing become more important in large catchments (Merz and Blöschl, 2009). Second, the catchment size closely interacts with the spatial variability of rainfall and runoff generation. Zhu et al. (2018) found that the spatial rainfall structure is more important for peak discharges in large catchments compared to small ones, partly because rainfall is generally less spatially variable in smaller catchments. In contrast, Wang et al. (2023) found that tail heaviness starts to increase at a lower degree of rainfall variability in small catchments than

in large ones, and concluded that small catchments are less resilient to the spatial variability of rainfall with respect to the emergence of heavy flood peak tails. With regards to runoff generation, Rogger et al. (2012) stated that the catchment storage tends to become more spatially variable with increasing catchment size. This makes the widespread simultaneous saturation in





a catchment and thus the occurrence of a step change in the FFC more likely in small catchments (Rogger et al., 2012), while at larger scales such pronounced non-linear behaviour of the runoff generation tends to be averaged out (Merz et al., 2022).

To improve the highly uncertain estimation of the upper-tail behaviour given the typical length of observations, a better understanding of the processes that control the tail behaviour and longer time series can be helpful. Both can be achieved through modelling approaches. Through a simulation-based approach, we can define and extract information about all relevant flood processes which lead to a certain tail behaviour. We can also simulate long time series to allow more robust statistical analyses. However, modelling approaches can only represent a simplified version of reality and cannot include all processes

relevant to flood generation in every detail. Using a modelling approach, Macdonald et al. (2024) analysed the effects of rainfall and runoff generation properties on flood peak tail behaviour. However, they focused on small homogeneous catchments and did not analyse the effects of spatial variability and catchment size.

In this study, we analyse how the spatial variability of rainfall affects the upper tail behaviour of flood peak distributions, and whether this effect depends on the tail of the rainfall distribution. We also investigate whether the spatial variability of

the runoff generation has an effect on flood peak tail behaviour. Finally, we analyse how those effects of spatially variable rainfall and runoff generation interact with catchment size, and whether there is a link between catchment size and the upper tail behaviour of flood peak distributions. To address these questions, we use a simulation model chain consisting of a weather generator, a rainfall–runoff model and a river routing routine. With this model chain, long time series of precipitation and discharge can be generated and their tail behaviour subsequently assessed for catchments of various sizes.

## 2  Methods

A simulation model chain is used to generate long time series of precipitation and discharge. The model chain consists of a stochastic weather generator, a conceptual rainfall-runoff model and a river routing routine (Fig. 1a). For the simulated precipitation and discharge time series, frequency analyses and an analysis of the respective upper tail behaviour are conducted. The simulations are run on a large synthetic catchment to allow a range of spatially variable setups. The different model setups

are defined according to the research questions: different degrees of spatial variability of rainfall and runoff generation are considered as well as rainfall distributions with varying tail behaviour (Fig. 1b), and the frequency analyses are conducted for catchments of various sizes. Overall, the climate conditions and catchment properties used in the model setups are characteristic for Central Europe.

### 2.1  Synthetic catchment

The synthetic catchment on which the simulations are run is generated using the R-Package OCNet (Carraro et al., 2020). The package is designed to generate and analyse optimal channel networks which reproduce "all scaling features characteristic of real, natural river networks" (Carraro et al., 2020). With a resolution of 2x2 $km^2$, a synthetic catchment is generated that has an area of 101,588 $km^2$ and elevations between 0 m and 805 m (see Fig. 2a). In the aggregation of the river network, we aim for a number of network nodes which is high enough to allow spatial analyses over a wide range of catchment sizes, while





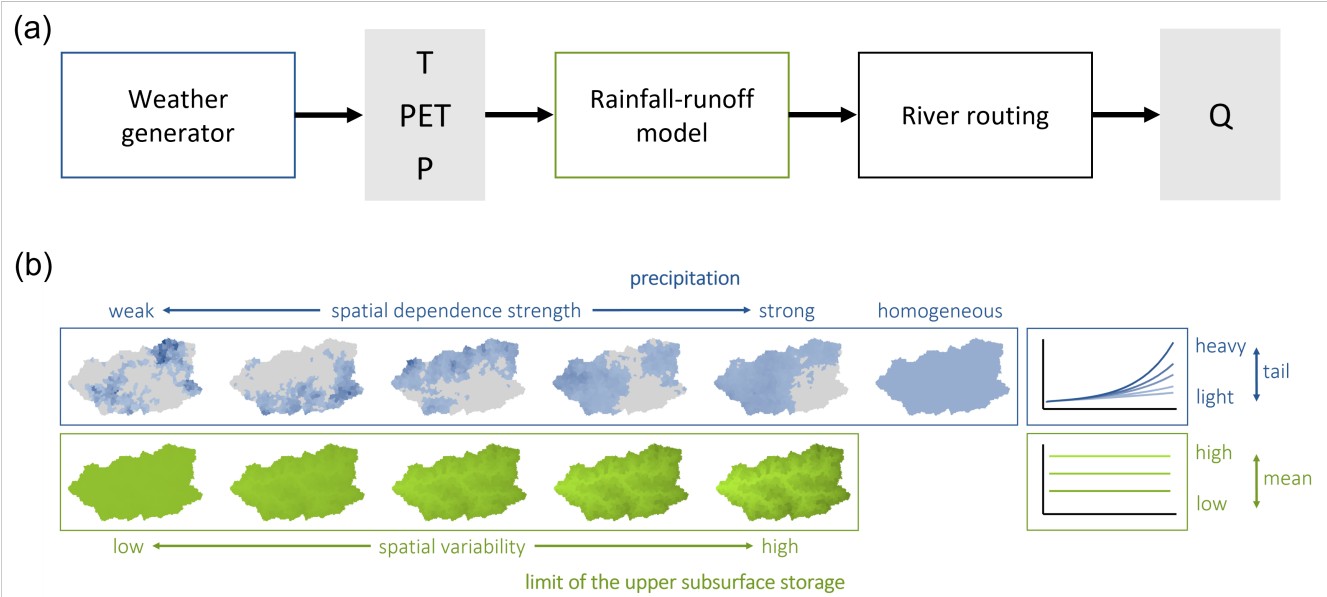

**Figure 1.** Schematics of the modelling approach. **(a)** The simulation model chain: Using a stochastic weather generator, time series of temperature (T), potential evapotranspiration (PET) and precipitation (P) are generated. These are then fed into a conceptual rainfall-runoff model coupled with a river routing routine to simulate long, continuous discharge time series. **(b)** The model setups: In the weather generator, different setups are generated by changing the spatial dependence strength and the tail behaviour of precipitation. In the rainfall-runoff model, the mean value and spatial variability of the limit of the upper subsurface storage are varied between setups.

balancing this with the increasing run time of the simulation model chain with increasing number of nodes. The final number of nodes is 678, corresponding to 678 sub-catchments in the catchment.

## 2.2 Simulation model chain

The first part of the simulation model chain is a stochastic multi-site, multi-variate weather generator. It is set up based on observational data from stations in Germany (Hundecha et al., 2009; Nguyen et al., 2021), and is used to generate time series of precipitation P, temperature T and potential evapotranspiration PET as input to the rainfall-runoff model. With regards to P, the weather generator has been evaluated to capture well both the daily mean and the extreme intensities for a large set of weather stations in Central Europe (Nguyen et al., 2021). Here, it is used to generate data for 678 locations in the synthetic catchment. The generated times series are based on observational data from the weather station in Bamberg (DWD, 2022), which is among the stations with the longest available records in Germany. For each configuration of the weather generator, 100 realizations of the same length as the observed record, i.e. 72 years, are generated with a daily resolution. The configurations differ in the tail behaviour of P and in the spatial variability of P. For the first aspect, the upper tail shape parameter of an extended generalized Pareto (extGP) distribution, which was fitted to the observed P data, is modified systematically. This is done by multiplying the





extGP shape parameter with a factor between 0.1 and 1.3. The lower tail shape and scale parameters of the extGP distribution remain as fitted to the data. The P time series with different tail behaviour are then re-scaled to have the same mean. To generate

P time series with different spatial variability, the spatial dependence strength of E-OBS data in Germany (Cornes et al., 2018) is derived and labelled as medium (M) dependence strength. Based on this, two weaker (weak W, medium weak MW) and two stronger (medium strong MS, strong S) dependence strength setups are derived. Note that these are "weak" and "strong" relative to the dependence strength derived for Germany, and not necessarily in absolute terms. The spatial dependence strength is used to derive data at all locations based on the Bamberg data. This approach and the manipulation of the extGP upper tail

shape parameter produce different spatial setups as well as time series with different frequency of extremes.

    The second part of the simulation model chain is a conceptual rainfall-runoff model following the structure of the HBV model (R-package TUWmodel; Parajka et al., 2007). It consists of a snow, a soil moisture, a response and a catchment routing routine with a total of 15 model parameters. The model is run on each of the 678 sub-catchments of the large synthetic catchment (101,588 km$^2$). It is forced by the time series of P, T and PET simulated with the weather generator as areal input.

The temporal specifications are the same as for the weather generator, i.e. 100 realizations of 72 years of daily data for each setup. However, the first 2 years of each realization and setup are used as warm-up period for the model and later removed from the analyses. 14 of the 15 model parameters are set to fixed values and do not vary between model setups. The values are based on the average parameter values of model calibrations for 273 Austrian catchments as reported by Merz et al. (2011), and can be found in Table A1 in Macdonald et al. (2024). The only model parameter that varies between model setups is the

storage capacity of the upper subsurface storage $L_{UZ}$. When this storage capacity is exceeded, an additional and faster runoff component $q_0$ is triggered. The spatial variability in $L_{UZ}$ thus affects whether saturation excess runoff is triggered locally or widely. In the model setups, three different mean levels of the storage capacity $\overline{L_{UZ}}$ are considered (23 mm, 43 mm, 63 mm) as well as different spatial configurations: for five degrees of variability, $L_{UZ}$ varies across space following the topography (Fig. 1b). The lowest storage capacity $L_{UZ,min}$ is assumed at the highest elevation $z_{max}$, and it increases linearly for lower

elevations:

$$L_{UZ,i} = \frac{L_{UZ,min} - \overline{L_{UZ}}}{z_{max} - \overline{z}} \cdot z_i + \frac{\overline{L_{UZ}} \cdot z_{max} - L_{UZ,min} \cdot \overline{z}}{z_{max} - \overline{z}} \tag{1}$$

    where $L_{UZ,i}$ is the storage capacity at the elevation $z_i$, and $\overline{z}$ is the mean elevation of the entire catchment. The five degrees of spatial variability are achieved by setting $L_{UZ,min}$ to different values between $\overline{L_{UZ}} - 2$ mm (lowest variability) and 1 mm (highest variability). The mean values and ranges of $L_{UZ}$ are defined based on the ranges reported in previous studies (Parajka

et al., 2007; Macdonald et al., 2024) and findings from test runs. The ratio of the mean rainfall depth to the catchment storage is relevant for the frequency of storage exceedances, and so $L_{UZ}$ is defined to cover a wide range of exceedance frequencies with the pre-defined rainfall data.

    The last part of the model chain is a river routing module, which routes the simulated runoff of each sub-catchment along the river network to the final outlet. The routine uses a cascade reservoir approach and is based on the Streamflow Synthesis

and Reservoir Regulation (SSARR) model (USACE, 1975). For the given catchment with a median length of river sub-reaches





of 10 km, the routing needs to be run on a sub-daily time scale to achieve accurate results. Through test runs, the optimum between accuracy and computational time is found to be a 2-hourly resolution. The daily runoff simulated with the rainfall-runoff model is disaggregated to 2-hourly values using linear interpolation between the simulated values, and re-aggregated to daily time series after the routing. The river routing is defined by three model parameters: the number of sub-reaches per river

reach (nbr), a coefficient affecting the time of storage per sub-reach (kts), and an exponential coefficient controlling the impact that a change in discharge has on the time of storage per sub-reach (n) (NOAA, 2003). The first one is commonly estimated using the characteristic reach length LC by taking the ratio between the total length of a river reach and LC (Pelin and Pahlsson, 2012). The three parameters are kept constant between all model runs and their values are based on physical considerations: the parameter values should result in realistic peak flows and travel times for a catchment of the given size. Previous studies

found travel times of flood peaks for 500 km long river reaches between 1.5 and 6 days (He, 2020; Allen et al., 2018; Meyer et al., 2018). Using the relation between catchment size and maximum observed daily peak flow of the 360 German catchments analysed by Macdonald et al. (2022), a peak flow of 4,500 $m^3$/s to 7,500 $m^3$/s is estimated for a catchment of 101,588 $km^2$. Travel times and peak flows within these ranges at the final outlet were used as criteria for defining the model parameters of the routing routine. The parameters are set to $L_C$ = 10 km, kts = 10, and n = 0.3. These values are within the ranges reported

in previous studies (Pelin and Pahlsson, 2012; USACE, 1975).

The output of each run of the model chain consists of the simulated discharge time series at each outlet, along with the time series of the share of the very fast runoff component and the respective model parameters and precipitation time series which were used in the model run. While we aimed to set up the model with climate conditions and catchment properties which are realistic for Central Europe, the simulated time series do not relate to real-world catchments. Instead, the focus is to analyse

the effects that specific changes in the model configurations have on the tail behaviour of flood peak distributions.

## 2.3 Analysis of the simulated time series

The simulated time series of P and Q and the spatial indices are analysed for catchments of different size. To this end, a subset of the 678 sub-catchments and their upstream catchments is selected as follows: first, a lower and an upper size limit are determined. The catchments should consist of at least 3 sub-catchments to allow spatial analyses, which results in a lower size

limit of 200 $km^2$. Further, there should be at least 3 distinct catchments of similar size to allow robust results, which leads to an upper size limit of 30,000 $km^2$. Second, nine catchment size classes within this range were defined with the following breaks: 200, 300, 500, 1,000, 2,000, 3,000, 5,000, 10,000, 20,000, and 30,000 $km^2$. Third, within each size class, all distinct catchments are selected. If there are nested catchments within one size class, only the largest one of them is considered. This results in 163 catchments (Fig. 2) for which all subsequent analyses are conducted.

To analyse the tail behaviour, annual maxima of the simulated P and Q time series are derived. For P, this is done for all 678 locations used in the weather generator. For Q, this is done only for the selected 163 outlets. After removing the warm-up periods, the 100 realizations of each model setup are combined to one long time series of 7,000 years for distribution fitting. GEV distributions are fitted to the annual maximum series (AMS) of P and Q using L-moments. To make the tail behaviour of



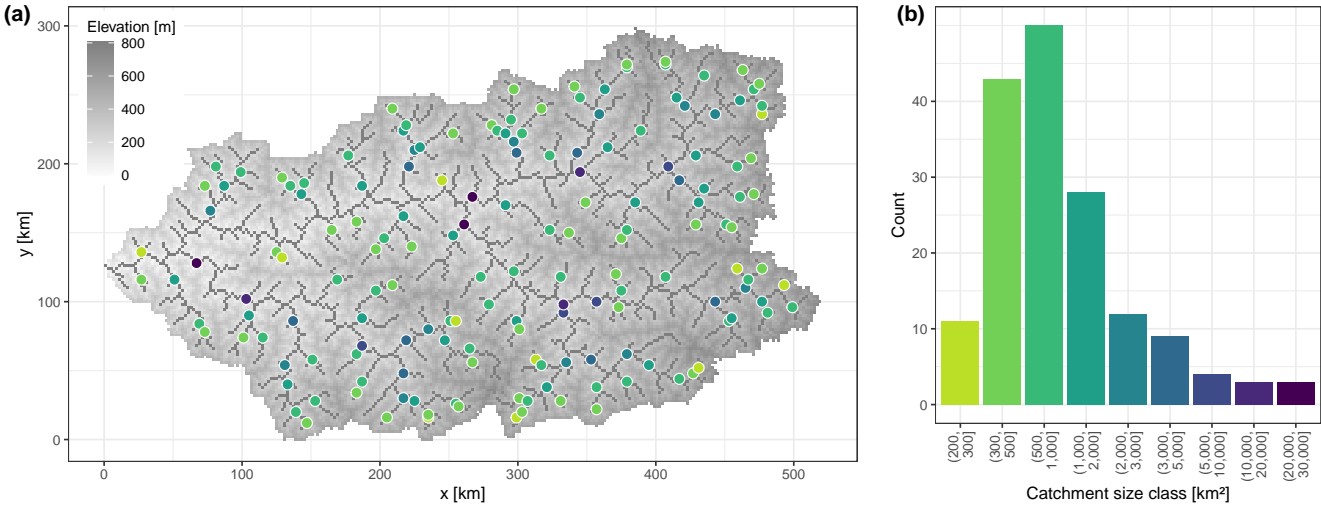

**Figure 2. (a)** The outlets of 163 catchments which are selected for analyses based on 9 size classes, and **(b)** the number of catchments per size class. Catchments within one class are not nested, while catchments across classes can be nested.

P and Q comparable, the shape parameters of the P distributions are aggregated to the same 163 outlets where Q is analysed:

for each outlet, the median shape parameter of the P distributions of all upstream sub-catchments is estimated.

The spatial variability of P is quantified for each model setup and for each of the 163 outlets. First, the spatial coefficient of variation $CV_{P,t}$ of P across all sub-catchments upstream of an outlet is estimated for each day $t$:

$$CV_{P,t} = \frac{\sqrt{\frac{1}{N}\sum_{n=1}^{N}(P_{t,n} - \overline{P_t})^2}}{\overline{P_t}} \tag{2}$$

where $N$ is the number of sub-catchments upstream of the respective outlet, $P_{t,n}$ is the rainfall depth in sub-catchment $n$ on

day $t$, and $\overline{P_t}$ is the mean rainfall depth across all $N$ sub-catchments on day $t$. Second, the median of $CV_{P,t}$ of all rainy days within the 7,000-year long time series is estimated ($CV_{P,med}$).

The spatial variability of the runoff generation is quantified for each model setup and for each of the 163 outlets. As a measure of spatial runoff variability, we consider how locally or widely saturation excess runoff is usually triggered in a catchment. For each day t, the number of sub-catchments upstream of an outlet where the very fast runoff component $q_0$ was

active is evaluated. The mean of this number is estimated for all days on which $q_0$ was active in at least one sub-catchment, and then divided by the total number of sub-catchments. This gives the average share of sub-catchments in which the very fast runoff component is triggered simultaneously, i.e. it is a measure of whether saturation excess runoff is usually triggered locally or widely in a catchment.



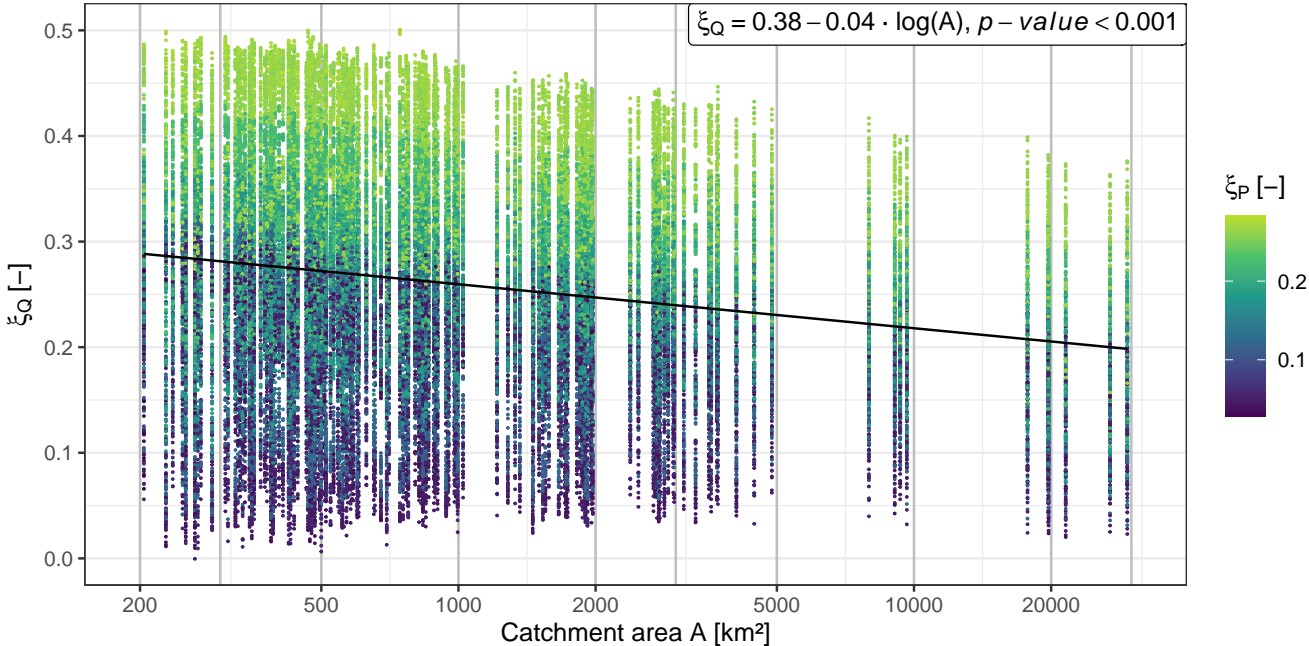

**Figure 3.** Shape parameters ($\xi_Q$) of Generalized Extreme Value (GEV) distributions fitted to simulated discharge series versus catchment area (A). Results are based on 375 model setups which are evaluated at 163 catchment outlets. The model setups differ in the tail behaviour ($\xi_P$) and spatial variability of the rainfall input, and in the mean value and spatial variability of the limit of the subsurface catchment storage. GEV distributions were fitted to annual maximum series of 7,000 years. A linear trend (black line) and its formula are displayed.

## 3 Results

Simulated discharge time series of 375 different model setups are analysed for 163 catchments ranging in size from 200 km$^2$ to 30,000 km$^2$. We find a clear downward trend in the GEV shape parameter of flood peak distributions with increasing catchment size (Fig. 3). While there is a large spread in the shape parameters between the different model setups for each catchment, there is a significant (p-value < 0.001) linear trend across catchment sizes. The GEV shape parameter decreases by 0.04 per order of magnitude of catchment size.

The catchment size is found to interact with the spatial variability of rainfall. For 25 different setups of the weather generator, which were evaluated at 163 catchments of various size, the spatial variability of P expressed as $CV_{P,med}$ is increasing with increasing catchment size (Fig. 4). This means that rainfall tends to be more variable in larger catchments. As expected, we also see a clear stratification of $CV_{P,med}$ with the spatial dependence strength specified in the respective setup of the weather generator. A weak spatial dependence strength of P results in more spatially variable rainfall and thus a higher $CV_{P,med}$ than

a strong spatial dependence strength.



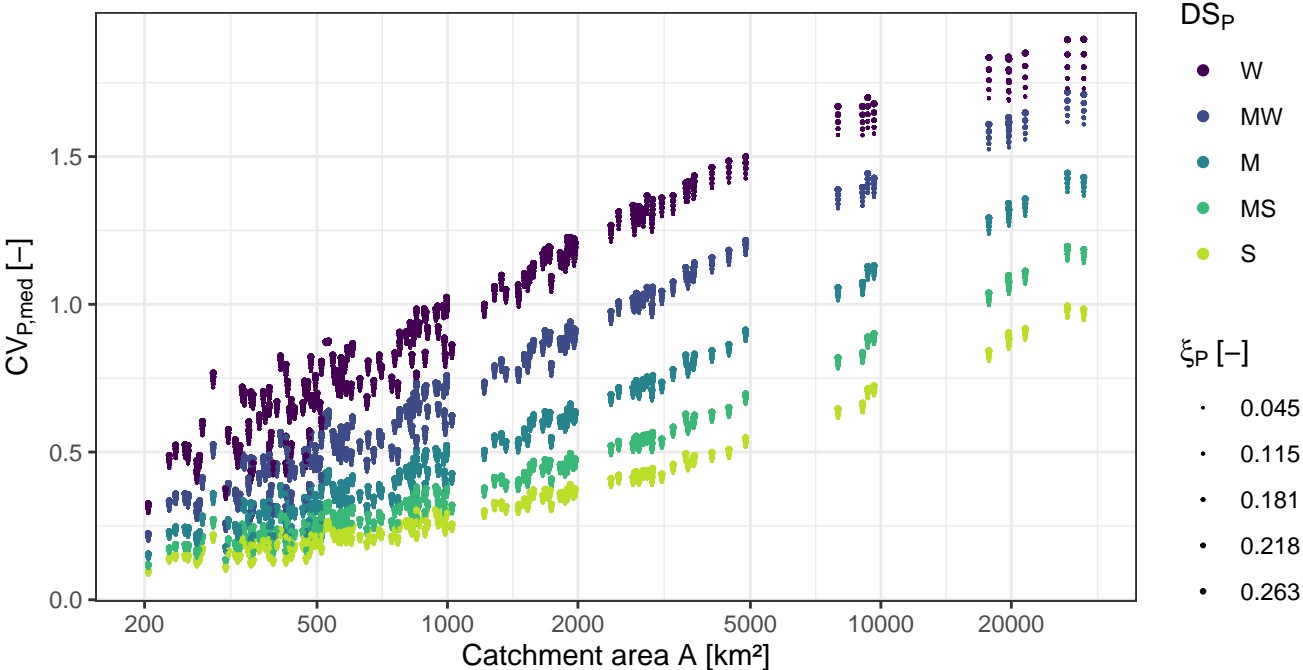

**Figure 4.** The median spatial coefficient of variation of precipitation ($CV_{P,med}$) versus catchment area (A) for 163 catchments. $CV_{P,med}$ is based on the daily rainfall in all sub-catchments of a catchment, and the median is taken across all rainy days of the 7,000-year long time series. Results are based on 25 setups of the weather generator: 5 values of spatial dependence strength $DS_P$ from weak (W) to strong (S), and 5 values of tail heaviness ($\xi_P$) of the underlying rainfall distribution.

When the GEV distribution of the forcing rainfall has a high shape parameter, the resulting discharge also tends to have a higher GEV shape parameter than that resulting from a rainfall with lighter tail (Fig. 3, Fig. 5). This is observed for different catchment sizes (Fig. 3) and for different degrees of spatial variability of the rainfall (Fig. 5).

Increasing spatial variability of rainfall tends to lead to a decrease of the flood peak shape parameter, at least when the spatial
variability of the catchment storage capacity is low (Fig. 5, left column). With increasing spatial variability of the catchment storage capacity, the downward trend becomes less clear. Especially when the spatial variability of rainfall is low, but that of the storage capacity is high (Fig. 5, right column), there is a large spread in the tail behaviour of flood peak distributions. In contrast, when the spatial variability of both rainfall and storage capacity is high, we see a downward trend in flood peak shape parameters against rainfall variability. This indicates that as rainfall variability increases it starts dominating the flood peak tail
behaviour independent of the degree of storage variability.

The overall downward trend in flood peak shape parameters with increasing rainfall variability should be considered with care, as we see a clear relation between the spatial variability of rainfall and catchment size (Fig. 4). As higher degrees of rainfall variability are often related to larger catchments, the observed trend might show the combined effect of rainfall variability and



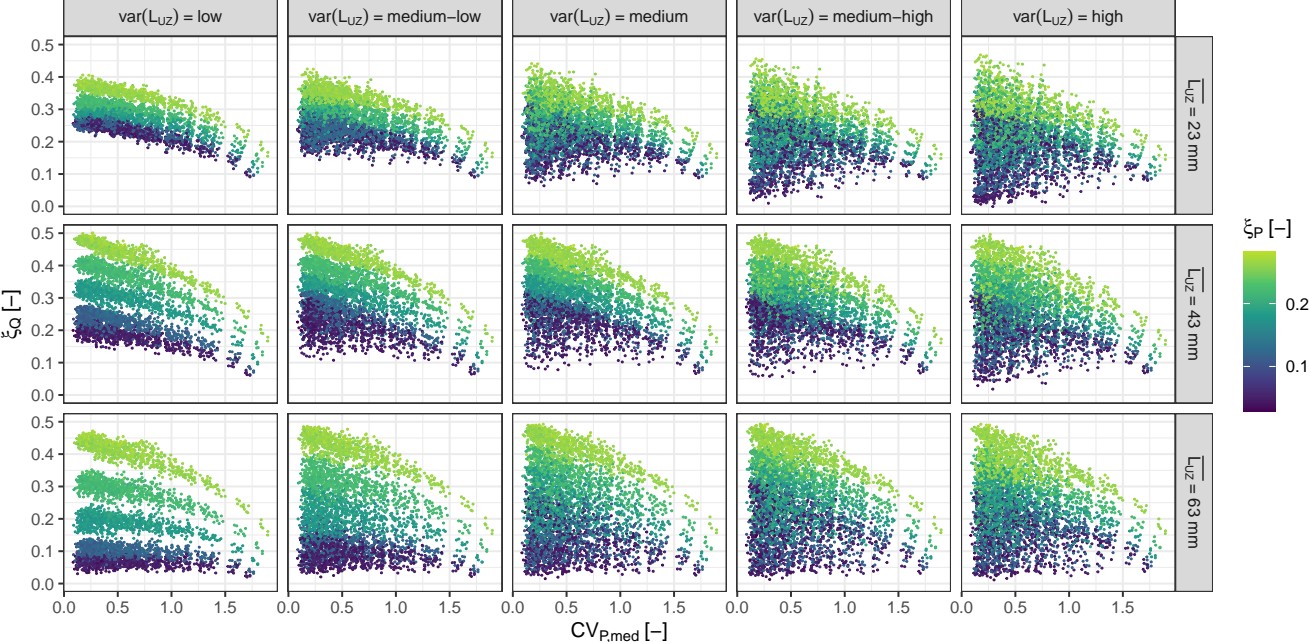

**Figure 5.** Shape parameters ($\xi_Q$) of Generalized Extreme Value (GEV) distributions fitted to simulated discharge series versus spatial rainfall variability expressed as the median spatial coefficient of variation of precipitation ($CV_{P,med}$). Results are based on 375 model setups which are evaluated at 163 catchment outlets. The model setups differ in the tail behaviour ($\xi_P$) and spatial variability ($CV_{P,med}$) of the rainfall input, and in the mean value ($\overline{L_{UZ}}$) and spatial variability (var($L_{UZ}$)) of the limit of the subsurface catchment storage. GEV distributions are fitted to annual maximum series of 7,000 years.

other processes that change with catchment size such as peak attenuation and river routing. To address this aspect, the effect of
the spatial variability of rainfall is analysed separately for two catchment size classes.

By considering only simulations with low variability of the catchment storage capacity and by evaluating size classes separately, we can reduce confounding effects. For the 54 smallest catchments (200 km$^2$ - 500 km$^2$) the rainfall variability $CV_{P,med}$ ranges between 0.085 and 0.83, while $CV_{P,med}$ is between 0.62 and 1.9 for the 10 largest catchments (5,000 km$^2$ - 30,000 km$^2$) (Fig. 6). There are fewer sub-catchments in the smaller catchments and therefore variability cannot get as high as in the larger
catchments. For the small catchments, there is a very slight or hardly any trend of flood peak shape parameters with increasing rainfall variability. For the large catchments, flood peak shape parameters decrease with increasing rainfall variability. Here, the downward trends are especially clear for high rainfall shape parameters and low catchment storage capacities. Further, we see an effect of the mean level of catchment storage capacity $\overline{L_{UZ}}$: for smaller catchment storages, the range of flood peak shape parameters for any given rainfall variability is smaller than for larger catchment storage capacities (Fig. 6). The larger
storage capacity seems to enhance the differences in flood peak shape parameters which are induced by different rainfall shape parameters.

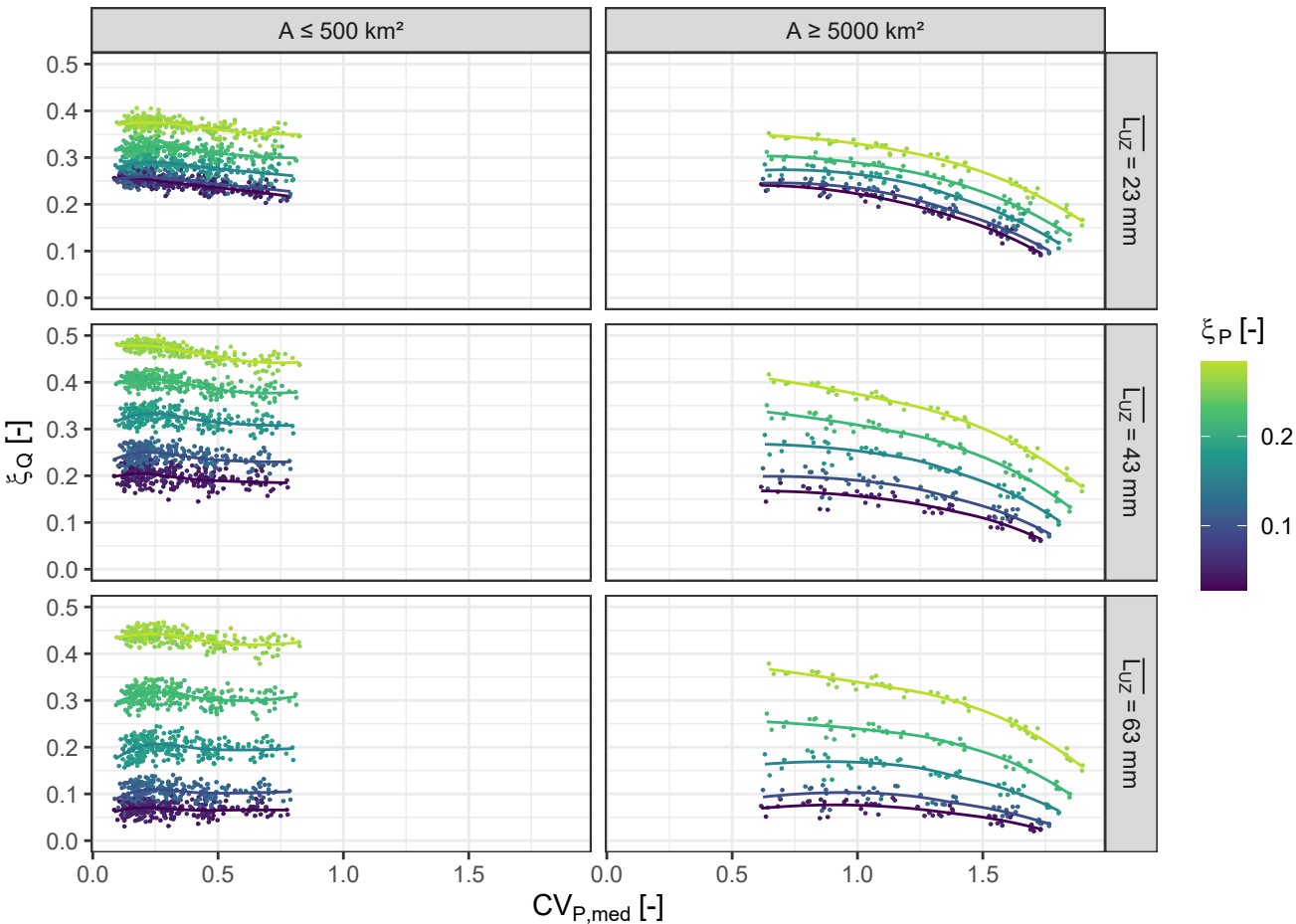

**Figure 6.** Shape parameters ($\xi_Q$) of Generalized Extreme Value (GEV) distributions fitted to simulated discharge series versus spatial rainfall variability expressed as the median spatial coefficient of variation of precipitation ($CV_{P,med}$). Results are based on 75 model setups with close to homogeneous catchment storage (i.e. var($L_{UZ}$) = low). The setups differ in the tail behaviour ($\xi_P$) and spatial variability ($CV_{P,med}$) of the rainfall input, and in the mean value of the limit of the subsurface catchment storage ($\overline{L_{UZ}}$). The simulations are evaluated for the 54 smallest and the 10 largest catchments. GEV distributions are fitted to annual maximum series of 7,000 years. Locally estimated scatterplot smoothing (LOESS) curves are fitted to subsets of simulations with the same rainfall tail behaviour.



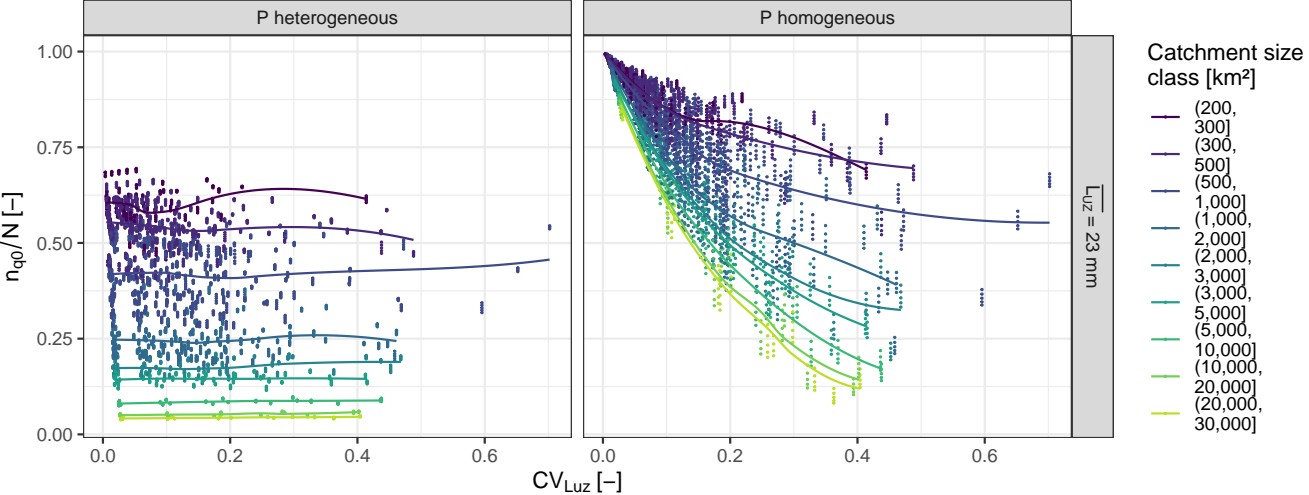

**Figure 7.** Average share of sub-catchments with simultaneous activation of the very fast runoff component ($n_{q0}/N$) against the spatial coefficient of variation of the catchment storage ($CV_{Luz}$). Results are based on 50 model runs which are evaluated at 163 catchment outlets. Model setups differ in the tail behaviour and spatial variability of the rainfall input, and in the spatial variability of the limit of the subsurface catchment storage ($L_{UZ}$). Heterogeneous precipitation (P) relates to the model setup with the weakest spatial dependence strength for P, while for homogeneous P the same time series of P is assumed for all sub-catchments. Locally estimated scatterplot smoothing (LOESS) curves are fitted for different catchment size classes.

To quantify the spatial variability of runoff generation, we consider whether saturation excess runoff is usually triggered locally or widely in a catchment. This is here characterized by the average share of sub-catchments in which the very fast runoff component $q_0$ is triggered simultaneously ($n_{q0}/N$). In analysing the effect of spatially variable runoff on flood peak

tail behaviour, we are particularly interested in the variability that is induced by catchment properties such as the storage capacity rather than by spatially variable rainfall. However, spatially variable rainfall has a strong effect on the spatial variability of runoff generation: for spatially variable rainfall, the share of sub-catchments in which the very fast runoff component is triggered simultaneously is not affected by the spatial variability of the catchment storage capacity (Fig. 7). On the other hand, for homogeneous rainfall we see a clear relation: the higher the spatial variability of the catchment storage capacity, the lower

is $n_{q0}/N$, meaning that saturation excess runoff occurs rather locally for very variable storage capacities. If we want to analyse the spatial variability of runoff generation that is related to catchment properties, we need to consider homogeneous rainfall to exclude strong confounding effects. However, assuming homogeneous rainfall conditions over a catchment of 101,588 km$^2$ is not very realistic. The following results should therefore not be mistaken as realistic simulations, while they can still provide insights into how certain processes affect the flood peak tail behaviour.

By considering only model runs with homogeneous rainfall, we can eliminate the effect of rainfall variability on saturation excess runoff, so that we see only the effect of runoff generation variability caused by spatial variability in the catchment storage capacity. Whether saturation excess runoff is triggered locally or widely has no clear effect on the tail behaviour of



**Figure 8.** Shape parameters ($\xi_Q$) of Generalized Extreme Value (GEV) distributions fitted to simulated discharge series versus the average share of sub-catchments with simultaneous activation of a very fast runoff component ($n_{q0}/N$). Results are based on 75 model setups with homogeneous rainfall, which are evaluated at 163 catchment outlets. Model setups differ in the tail behaviour ($\xi_P$) of the rainfall input, and in the mean value ($\overline{L_{UZ}}$) and spatial variability of the limit of the subsurface catchment storage. GEV distributions are fitted to annual maximum series of 7,000 years.

flood peak distributions (Fig. 8). When saturation excess runoff is usually triggered very widely in a catchment, i.e. when the average share of sub-catchments with simultaneous activation of the very fast runoff component is close to 1, the GEV shape

parameters are clustered around a few values – those clusters differ in terms of the tail behaviour of the rainfall distribution and in the mean storage depth. For spatially variable runoff generation, i.e. when saturation excess is usually triggered more locally, we see point clouds with no clear trends or relations. These point clouds look however different for the different mean levels of the catchment storage capacity.





There appears to be a nonlinear relationship between the shape parameters of the flood peak distributions and the average depth of the subsurface catchment storage (Fig. 9). The flood peak shape parameters are low for small and large catchment storage capacities and have a maximum for medium storages. This nonlinear behaviour seems to be related to the distribution fitting rather than to hydrological processes: when looking at the frequency curves of simulated annual maxima and the respective fitted distributions (see Fig. 10 for examples), the tail behaviour of the simulated peak discharges is not always well represented by the fitted GEV distribution. There can be a step change in the annual maxima, and the return period of the step change affects how well the fitted distribution can represent the data. The return period in turn depends on rainfall characteristics and the catchment storage. For example, in cases with small catchment storage capacities and light-tailed precipitation, the fitted GEV distributions overestimate the flood peaks with high return periods (Fig. 10, top row).

In the setup with the least variable catchment storage capacities, the estimated mean storages for all 163 catchments cluster around the mean values used in the model setups, i.e. 23 mm, 43 mm, and 63 mm (Fig. 9). In contrast, for the most variable setup of catchment storage capacities, the estimated mean storages range from 4 mm to 102 mm. The mean values are fixed in the model setup for the entire large catchment but we analyse sub-catchments and they can have very different mean storages depending on where in the large catchment they are located.

How locally or widely saturation excess runoff is usually triggered in a catchment has only a very little effect on the flood peak shape parameter compared to the mean catchment storage (Fig. 9). For heavy-tailed rainfall distributions, a more variable runoff generation at the same level of mean storage capacity tends to lead to a lower flood peak shape parameter. For light-tailed rainfall distributions, this is only true for medium catchment storage capacities while for larger storage capacities a more variable runoff generation leads to higher shape parameters.

## 4 Discussion

The trend we found of decreasing flood peak shape parameters with increasing catchment size is in line with some previous studies (e.g. Macdonald et al., 2022; Villarini and Smith, 2010). Other studies did not find this trend (e.g. Smith et al., 2018; Morrison and Smith, 2002). The stream gauges analysed by Morrison and Smith (2002) are most likely also included in the larger set of gauging stations analysed by Villarini and Smith (2010), and the latter ones analysed much longer times series – at least 75 years of observations compared to 30 years for Morrison and Smith (2002). The trend found by Villarini and Smith (2010) therefore appears more reliable. They found a decrease of the shape parameter of 0.07 per order of magnitude of the catchment size for 572 catchments in the eastern United States, which is slightly stronger than the decrease of 0.04 resulting from our analysis. This could be a regional difference, or related to the modelling setup: some studies show that rainfall tails become lighter for larger areas (Merz et al., 2022; Overeem et al., 2010; Dyrrdal et al., 2016), but in our setup we keep the rainfall tail constant across catchment scales. Adding this effect could enhance the downward trend of flood peak shape parameters against catchment sizes. A potential explanation for the trend is that in small catchments, distinct non-linear behaviours in the runoff generation and convective rainfall bursts can result in heavy tails, while in larger catchments, where routing effects become increasingly important, these effects might be averaged out (Merz and Blöschl, 2009).

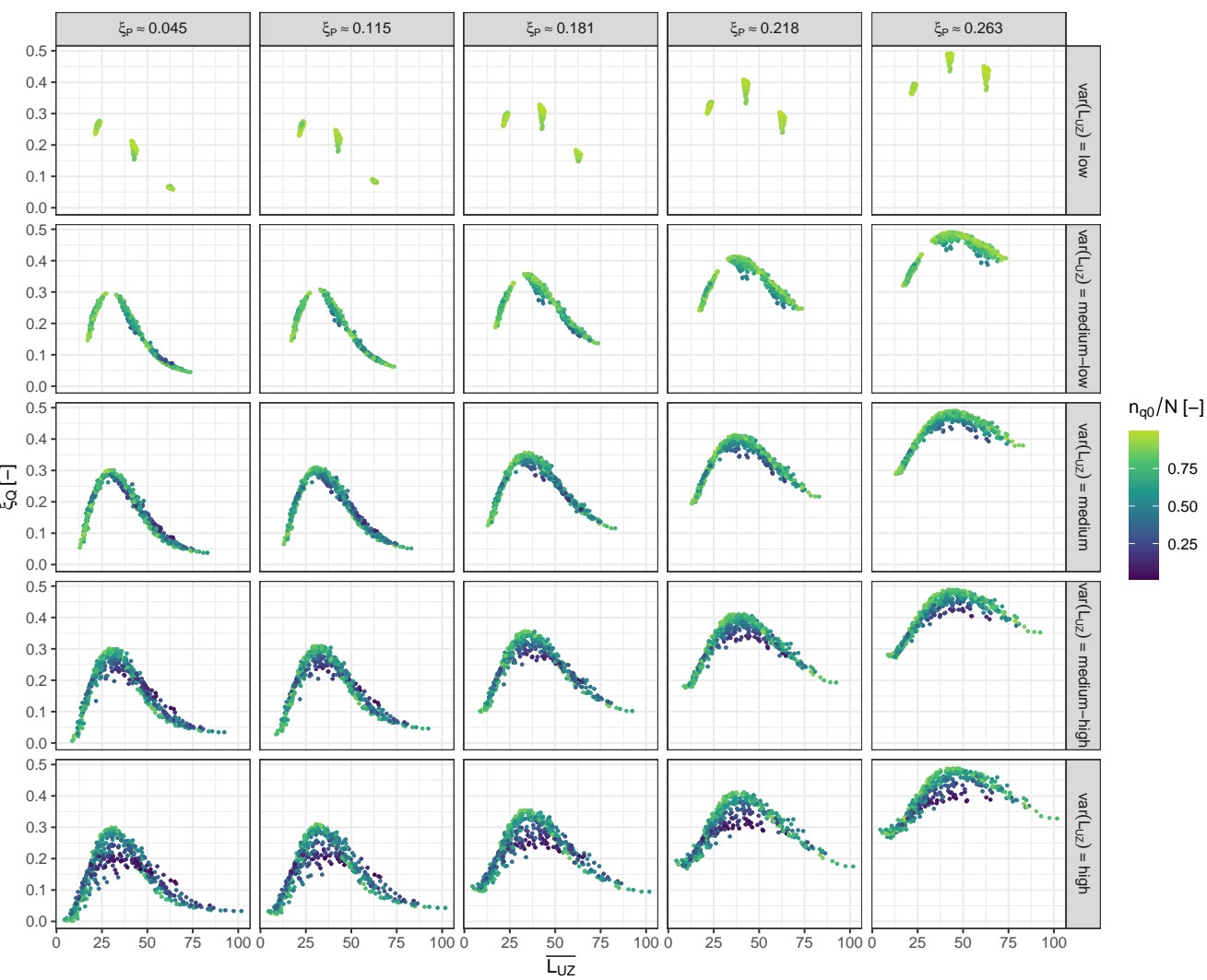

**Figure 9.** Shape parameters ($\xi_Q$) of Generalized Extreme Value (GEV) distributions fitted to simulated discharge series versus the mean value of the limit of the subsurface catchment storage ($\overline{L_{UZ}}$). Results are based on 75 model setups with homogeneous rainfall, which are evaluated at 163 catchment outlets. Model setups differ in the tail behaviour ($\xi_P$) of the rainfall input, and in the mean value and spatial variability of the limit of the subsurface catchment storage. The latter results in spatial variability of runoff generation which is here expressed as the average share of sub-catchments with simultaneous activation of a very fast runoff component ($n_{q0}/N$). GEV distributions are fitted to annual maximum series of 7,000 years.



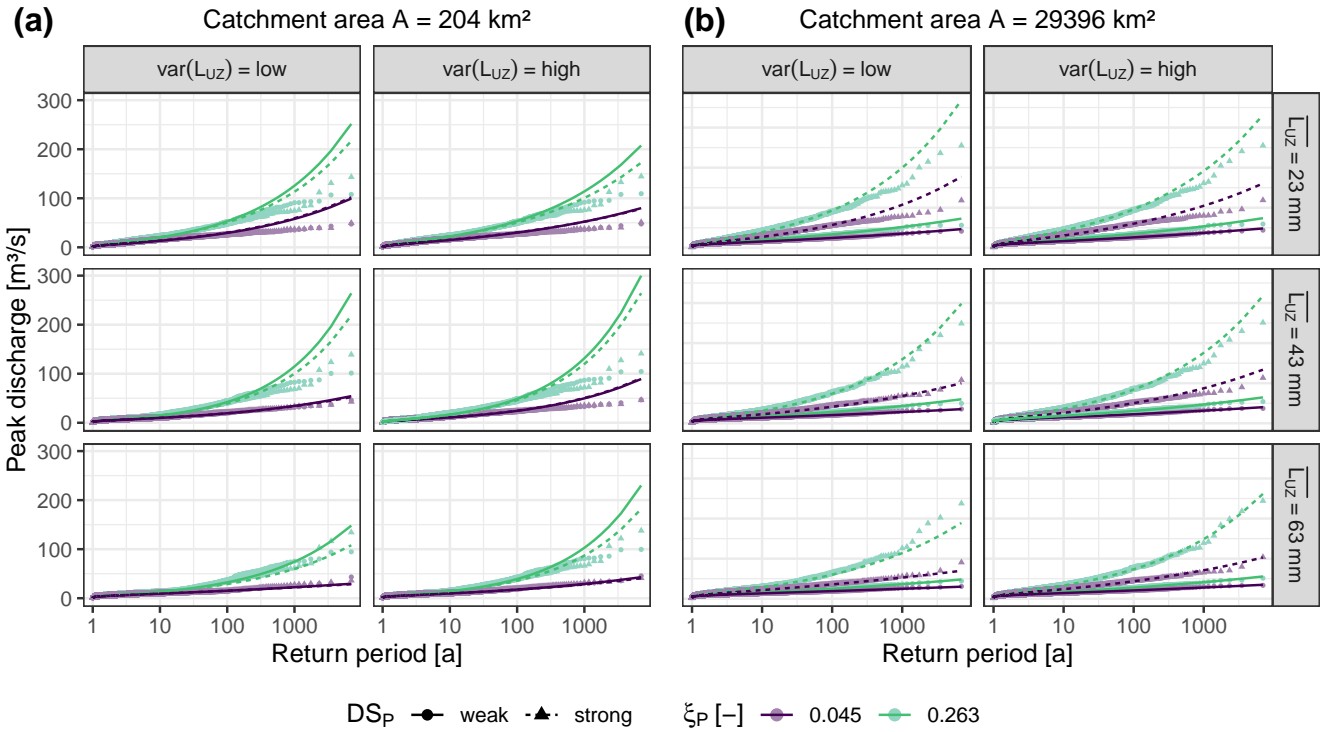

**Figure 10.** Example flood frequency curves (FFCs) for **(a)** the smallest and **(b)** the largest catchments analysed. The annual maxima of the simulated discharge series of 7,000 years are presented (dots and triangles) along with the Generalized Extreme Value (GEV) distributions fitted to those annual maxima (solid and dashed lines). The FFCs result from 16 model setups which differ in the tail behaviour ($\xi_P$) and spatial dependence strength (DSP) of the rainfall input, and in the mean value ($\overline{L_{UZ}}$) and spatial variability (var($L_{UZ}$)) of the limit of the subsurface catchment storage.

Irrespective of the degree of spatial variability of rainfall, we found that a higher shape parameter of the rainfall distribution tends to lead to a higher shape parameter of the flood peak distribution. This is in line with findings of Macdonald et al. (2024) for small homogeneous catchments and with Gaume (2006) who stated that rainfall statistical properties asymptotically

control the shape of flood peak distributions. We estimate the rainfall shape parameter for each catchment by taking the median shape parameter of the P distributions of all sub-catchments. An alternative approach would be to aggregate P on a daily basis across the catchment area and then derive the annual maxima and shape parameter from the aggregated, areal P. We assume that the slightly simpler approach that we use gives adequate results for our setup, as we do not vary the rainfall tail behaviour in space. In each setup of the weather generator, rainfall is generated with a fixed upper tail shape parameter of the

extGP distribution across all sub-catchments. If the rainfall tail behaviour would vary strongly in space within a catchment, the alternative approach is deemed more appropriate for estimating rainfall tail behaviour that is representative of the entire catchment.



We found a decreasing trend of flood peak shape parameters with increasing spatial variability of rainfall, especially for large catchments, and no clear trend for small catchments. For small catchments, the range of spatial variability considered
was smaller than for the larger catchments, as the spatial variability is limited by the number of sub-catchments in a catchment in our setup. The decreasing trend that we found for large catchments seems to oppose the results of Wang et al. (2023), who found that increasing spatial variability of rainfall leads to heavier tails of flow distributions beyond a certain degree of variability. They based this finding on scenario simulations for five German catchments (98 km$^2$ – 2,841 km$^2$). In 2 of their 3 scenarios, they kept the spatial variability of rainfall fixed in time, i.e. for all precipitation events and resulting floods. The
spatial coefficient of variation of rainfall $CV_P$ in their setups ranged from close to zero to well beyond 10 (i.e. 1000 %), and the degree of variability beyond which they found an increase in tail behaviour was a $CV_P$ of 2 or larger. The daily $CV_P$ values in our study cover a similar range (Fig. 11), while the median across all rainy days in a catchment is always below 2. The rainfall variability in our setups is based on the spatial dependence strength estimated for E-OBS rainfall data in Germany, with two levels of weaker and two levels of stronger dependence strength than observed. The range of rainfall variability in
Wang et al. (2023) is based on the estimated spatial variability for 175 German catchments based on REGNIE data. The main difference seems to be that Wang et al. (2023) consider $CV_P$ to be constant in time. They take a value above the 95th percentile of daily $CV_P$ values as the upper bound of the spatial variability in their simulation setups – this means that values which were observed on less than 5 % of the days in all catchments are used in some setups to define the spatial variability on every day, i.e. constant in time. Although spatial variability of rainfall might increase in the future due to an increase in intensity and a
decrease in spatial extent (e.g. Wasko et al., 2016; Peleg et al., 2018), it seems unlikely that such high degrees of variability will become persistent in time. We believe that this rather unrealistic assumption of Wang et al. (2023) is the main cause for finding opposing trends in flood peak tail behaviour with increasing spatial variability.

We explain the decreasing trend in tail-heaviness with increasing spatial variability that we found for large catchments as follows: in the least variable setup, extreme rainfall events occur simultaneously in a large share of the catchment and can
therefore lead to widespread enhanced runoff generation which in turn can result in a distinctively higher flood peak at the outlet of the catchment. In a more variable setup on the other hand, extreme rainfall events occur only localised and therefore also enhanced runoff generation might be triggered only locally. Such a local flood peak is then attenuated on its way to the catchment outlet. In small catchments, we did not find a clear trend of flood peak shape parameters against rainfall variability and the likely reason for this is the smaller attenuation effect in smaller catchments.

The spatial variability of runoff generation caused by spatially variable subsurface catchment storage capacities does not show a clear effect on flood peak tail behaviour. When the average share of sub-catchments with simultaneous activation of the very fast runoff component is close to 1, saturation excess runoff is usually triggered widely in a catchment. In this case, the flood peak shape parameters are clustered around a few values (Fig. 8) and these clusters vary according to the tail behaviour of the rainfall distribution and the mean storage depth. This influence of the rainfall tail behaviour and the mean
storage depth for near-homogeneous runoff generation can be compared to the findings from Macdonald et al. (2024). For small catchments with homogeneous rainfall and runoff generation, they found that beyond a certain return period, the tail of the rainfall distribution asymptotically controls the tail of the flood peak distribution, and that this return period depends on

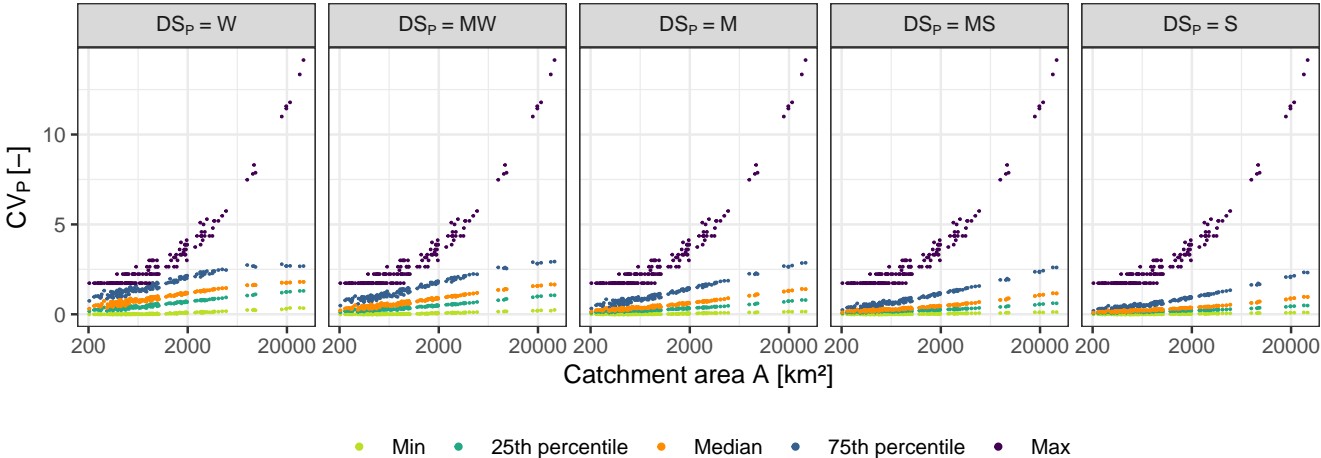

**Figure 11.** Percentiles of the spatial coefficient of variation of precipitation ($CV_P$) versus the catchment area (A) of 163 catchments. $CV_P$ is based on the daily rainfall in all sub-catchments of a catchment, and for each catchment the percentiles across all rainy days of the 7,000-year long time series are presented. Results are based on 5 setups of a weather generator with weak (W) to strong (S) spatial dependence strength DS$_P$ between rainfall depths in the sub-catchments.

the ratio of catchment storage to catchment wetness. We assume the storage capacity to follow the topography, with shallow storages at high elevations and deeper storages at lower elevations as soil depth is usually decreasing with increasing elevation

(Scull et al., 2003; Van Tol et al., 2013). Test runs indicated though that the exact spatial pattern in the spatially variable setups do not affect the findings (not shown).

In contrast to its spatial variability, the average depth of the subsurface catchment storage appears to affect the shape parameter of flood peak distributions in a nonlinear way. If the storage capacity is exceeded only by large but not by smaller rainfall events, this can result in a step change in the flood frequency curve (Rogger et al., 2012). The size of the storage capacity

determines the return period at which the step change occurs (Rogger et al., 2012). This means that the average depth of the subsurface storage is relevant for how often the storage capacity is exceeded for a given rainfall level. Spatial variability in the storage capacity or soil depth has been found to smooth out step changes in the FFC (Rogger et al., 2013; Struthers and Sivapalan, 2007). Our results indicate that the return period at which a step change occurs in the FFC has a stronger effect on the tail behaviour of the fitted distribution than how pronounced this step change is. However, as Fig. 10 shows, having a

step change in the FFC can mean that the annual maxima are not well represented by a GEV distribution. For example, in the model setups with low storage capacity and light-tailed rainfall, some simulated annual maximum flood peaks show a kind of S-shape with a step change at a low return period and a confined increase in flood peaks with increasing return periods beyond this. Such a shape of the FFC cannot be represented by a GEV distribution and so the fitted distributions overestimate the flood peaks with high return periods. This shows that GEV distributions can have a poor fit when a process shift is present.

Nevertheless, GEV distributions are commonly used in hydrological practice, for example, for the estimation of design floods.





In conclusion, the pattern of the GEV shape parameter versus the average storage capacity that we found is most likely linked to the distribution fitting and only indirectly to hydrological processes.

The findings from this study are based on synthetic catchments and simulation runs. This has the great advantage that much longer time series can be generated than are usually available from observations. This allows a more robust estimation of the

tail behaviour as the sampling uncertainty of the GEV shape parameter decreases with increasing sample size (Wietzke et al., 2020). However, results from simulation runs are first of all representative for the simplified world represented by the models and are not necessarily transferable to the real world. For example, in our setup, we only considered how locally or widely saturation excess runoff is triggered to represent the spatial variability of runoff generation. In a real catchment, multiple flow paths and runoff components might be activated during extreme events. As all models, the rainfall-runoff model that we use

is a simplified representation of reality and is not able to represent all potential runoff generation processes. Nevertheless, the findings based on simulated time series can still give us valuable insights into how the analysed processes affect the tail behaviour of flood peak distributions. This is particularly the case as the different parts of the simulation model chain have been shown to represent well real-world behaviour when calibrated with real-world data (e.g. Nguyen et al., 2021; Ceola et al., 2015; Parajka et al., 2007).

## 5   Conclusions

Rainfall characteristics and runoff generation processes can affect the tail behaviour of flood peak distributions. Here, we analysed how the spatial variability of rainfall and runoff generation influence flood peak tail behaviour, and whether and how this interacts with the size of catchments. To address these questions, a simulation-based approach was used: a model chain consisting of a weather generator, a rainfall-runoff model and a river routing routine was set up for a large, synthetic catchment.

Different configurations of the models were designed to represent varying degrees of spatial variability of rainfall, varying tail behaviours of the rainfall distributions, varying mean catchment storage depths and varying degrees of spatial variability in the runoff generation. With these setups, 7,000 years of discharge were simulated, generalized extreme value (GEV) distributions fitted, and their tail behaviour analysed. This was done for 163 catchments ranging between 200 km$^2$ and 30,000 km$^2$ in size.

We found that the GEV shape parameter decreases with increasing catchment size, meaning that smaller catchments tend

to have flood peak distributions with heavier tails. Independent of the catchment size, a rainfall distribution with a heavier tail results in a flood peak distribution with a heavier tail. Further, the shape parameter of flood peak distributions was found to decrease with increasing spatial variability of the rainfall, especially for large catchments. This is most likely linked to the flow attenuation effects in large catchments through which local flood peaks are balanced out on their way to the catchment outlet. With regards to runoff generation, we found no clear effect on the flood peak tail behaviour depending on whether saturation

excess runoff usually occurs locally or widely in a catchment. In contrast, the average depth of the catchment storage seems to have a nonlinear effect on the GEV shape parameter of flood peak distributions. These results suggest that how frequently saturation excess runoff is triggered has a stronger effect on the tail behaviour of flood peak distributions than how locally or widely this happens. However, the identified effect of the mean storage capacity on the GEV shape parameter might be to some

degree related to aspects regarding the distribution fitting. When process shifts are present in a catchment, the flood frequency
curve might show a step change, and as a result the flood peaks might not be represented well by a GEV distribution. This
should always be kept in mind when using GEV distributions for the estimation of design floods.

Overall, the spatial variability of rainfall shows a much stronger effect on the tail behaviour of flood peak distributions than
the spatial variability of the runoff generation. The effect of spatially variable rainfall is closely interlinked with the catchment
size, and attenuating effects in large catchments are assumed to lead to lighter tails. The findings are based on simulation runs
so that future studies are required to validate the findings in real-world catchments.

*Code and data availability.*    The code of the regional weather generator is available in a GitLab repository (https://git.gfz-potsdam.de/hydro/
rfm/rwg, last access: 30 January 2023). Access can be granted by Dung Viet Nguyen upon request. The rainfall–runoff model TUW-model is
available as an R package (https://CRAN.R-project.org/package=TUWmodel, Viglione and Parajka, 2020). The observational data from the
weather station in Bamberg can be obtained from the Climate Data Centre of the Deutsche Wetterdienst (https://opendata.dwd.de/climate_
environment/CDC/observations_germany/climate/, DWD, 2022).

*Author contributions.*    BM, SV and EM conceptualized the study. BM and SV supervised the study. EM performed the simulations and
formal analysis with contributions from DN. EM prepared the manuscript with contributions from all co-authors.

*Competing interests.*    The authors declare that they have no conflict of interest.

*Acknowledgements.*    The financial support of the German Research Foundation (Deutsche Forschungsgemeinschaft, DFG) for the research
group FOR 2416 "Space-Time Dynamics of Extreme Floods (SPATE)" is gratefully acknowledged. Viet Dung Nguyen was funded by the
Federal Ministry of Education and Research of Germany in the framework of the project FLOOD as a part of the ClimXtreme Research
Network on Climate Change and Extreme Events.





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
