# Peer review of "Heavy-tailed flood peak distributions: What is the effect of the spatial variability of rainfall and runoff generation?"

_Hydrology and Earth System Sciences, 2024_

## Author Response (AR1)

Dear Reviewers, dear Editors,

Thanks a lot for the thorough and constructive reviews and useful comments! We have tried to improve the original manuscript accordingly, following the reviewers' suggestions closely. Please find our responses to all comments below (in *italic*). Line numbers in our responses refer to the revised version of the manuscript.

**'Referee Comment on hess-2024-181', Anonymous Referee #1**

I enjoyed reading this paper, which nicely follows other works of the same Authors on understanding why and how are flood frequency curves heavy tailed. With a simulation approach, the authors shed light on one interesting question that is still unclear in flood research: In which way does the spatial variability of rainfall and runoff generation affect the shape of the tail of flood distribution? The paper is nicely written and clear with an interesting discussion on how the results compare with existing literature. I just have a few minor comments:

*We thank the reviewer for the thorough review, their appraisal and comments. We will address them one by one below and state what changes we made to the manuscript.*

- The GEV model is fitted to the simulations using the L-moments method. Since timeseries are so long (7000 years) maybe other methods could be better, for example the method of maximum likelihood perhaps giving more weight to the tail of the distribution (not sure how but it should be possible) and so reducing the effect of step changes in adapting a GEV to the tail of the distribution. In other words, my concern is that the fitting method used here puts a lot of weight on small floods, which are not really of interest.

*It is true that the L-moment fitting puts a bit more weight on small floods than the maximum likelihood method (MLM) does. However, MLM is a lot more computationally expensive due to the need of iterative procedures. For some test cases, we found a factor of 30-40 between the times needed for fitting a distribution with either of these two methods. This might be negligible when fitting distributions to a few short time series, but really adds up when hundreds of time series of thousands of years are analysed. Further, previous studies comparing the asymptotic standard errors of L-moment fitting and MLM found that that L-moment method is reasonably efficient and sometimes even more efficient in parameter estimation than MLM (Hosking, 1990; Hosking and Wallis, 1987). In addition to this, a very recent study by Vogel et al. (2024) highlights the advantages of using L-moments when working with heavy-tailed data, as L-moments are always finite when the mean exists even if higher moments do not exist. Based on these considerations, we decided to use the L-moments method. We added in the manuscript (l. 214):*

*"L-moment fitting has been evaluated to be reasonably efficient in parameter estimation compared to the maximum likelihood method (Hosking, 1990), while being a lot less computationally expensive. In addition to this, a recent study by Vogel et al. (2024) highlights the advantages of using L-moments when working with heavy-tailed data, as L-moments are always finite when the mean exists even if higher moments do not exist."*

- Since it is a simulation experiment and data are not a problem, why is the daily resolution used and not hourly, for example? The spatial resolution is quite high (2x2 km) and the smaller subcatchments may have short concentration times.

*It is true that data is less of a problem in simulation-based studies, however we still need it for setting up the model. Using an hourly resolution would have drastically limited the data available for setting up and calibrating the weather generator. In addition, we are limited by computational resources. Computational costs would have increased a lot when increasing the temporal scale for the rainfall-*

*runoff model but even more so for the weather generator. Running the weather generator on the given catchment size with the given spatial resolution, for the given number of model setups and the given length of time series (7000 years) but with an hourly resolution simply was not feasible. To balance between accuracy of the simulations and computational costs/feasibility, we decided to use the daily resolution for the weather generator and the rainfall-runoff model. For the river routing on the other hand, a daily resolution would have resulted in very inaccurate or partly even unrealistic results. Therefore, we decided to disaggregate the data to a 2-hourly resolution (providing the same accuracy as hourly at lower computational costs) for this part of the model chain (see p. 6, l. 177). We added in the discussion (l. 405):*

*"For some small catchments, running the model at an hourly instead of a daily resolution would have been more appropriate as concentration times might be sub-daily. However, this would have increased the computational costs drastically. Especially for the weather generator, generating time series of 7000 years for the given setups in an hourly resolution did not seem feasible. Further, a higher temporal resolution would have limited the data available for setting up and calibrating the weather generator. With the aim of balancing between accuracy and computational feasibility, the weather generator and rainfall-runoff model were run at a daily scale, while the data was disaggregated to a higher temporal resolution for the river routing."*

- The models are described in other papers but maybe it would help the reader to have some more details. For instance the routing model seems to be quite simple, with the 3 parameters described at page 7, and equations could be given in this paper.

*Thanks for this suggestion. We added some details on the part of the rainfall-runoff model which is most relevant to our study, and on the river routing. We made additions in l. 160 as follows:*

*"When this storage capacity is exceeded, an additional and faster runoff component q0 is triggered:*

$$q_0 = \frac{S_{UZ} - L_{UZ}}{k_0}$$

*where Suz is the storage in the upper zone and k0 is the storage coefficient for the very fast runoff response."*

*We further changed l. 183:*

*"The river routing is defined by three model parameters: the number of sub-reaches per river reach (nbr), a coefficient affecting the time of storage per sub-reach (kts), and an exponential coefficient controlling the impact that a change in discharge has on the time of storage per sub-reach (n) (NOAA, 2003). The time of storage TS per sub-reach is defined as*

$$TS = \frac{kts}{Q^n}$$

*with Q being the discharge."*

- The first result, i.e., that more spatial variability of rainfall events results in weaker tail heaviness in large catchments, and therefore in lower possibility of surprises, is perhaps good news (for large catchments) given that climate change seems to be leading to more intense but more localized storm events. Maybe some discussion on the implications of the results in climate change research would be useful.

*Thanks for drawing our attention to this interesting aspect. We added the following paragraph to the discussion (l. 368):*

*"Finding lighter flood peak tails for increasing spatial variability of rainfall in large catchments could be seen as good news in the light of climate change. The spatial variability of rainfall is expected to increase in a warmer climate due to an increase in intensity and a decrease in spatial extent, as studies on convective rainfall and storm cells found (Wasko et al., 2016; Peleg et al., 2018). Such spatially more concentrated precipitation can be interpreted as increased spatial variability at the scale of large catchments, and might therefore reduce the tail heaviness and thus the chance of surprising floods. However, the situation looks very different for small catchments: even more localised storm cells can cover the entire catchment area and the prospective intensified rainfall would result in more severe flooding. A shift to more spatially variable rainfall in a warmer future should therefore not necessarily be taken as good news for large catchments, but rather indicates that the analysis of the flood peak tail behaviour will become increasingly important for small catchments."*

- The second result is that the spatial variability of runoff generation, although having an effect on the shape of step changes in flood frequency curves, has a weak effect on the shape of the tail of the distribution (as measured by the GEV shape parameter). Instead, the average depth of catchment storage, and therefore the position of the step change in flood frequency curves, has an effect on the tail heaviness (but this may be due to the way this heaviness is measured, i.e., by fitting a GEV distribution that may not be the best choice for S shaped frequency curves). Are there practical implications of these results, i.e., for hydrological modelling or flood design?

*Some considerations on the implications of these findings for practice are as follows: If tail heaviness could be directly linked to the average depth of catchment storage, this would allow enhanced tail estimations based on good knowledge of the storage of the respective catchment. In this case, studying time-invariant characteristics of a catchment of interest in detail could result in more accurate tail estimations. However, as mentioned, the observed effect might be due to issues regarding the distribution fitting. This in turn suggests that further studies on suitable probability distributions for annual maximum floods would be valuable. Overall, these aspects should be analysed in more detail, and currently we do not feel confident to give specific recommendations for hydrological practice based on this finding.*

- Line 99: the sentence "However, modelling approaches can only represent a simplified version of reality and cannot include all processes relevant to flood generation in every detail" is true but I would have expected a following sentence stating that, still, a lot can be learned through modelling approaches, since that's the approach used in the paper.

*Thanks for pointing this out. A similar statement can be found in the discussion (l. 418), but we also added a shorter version here:*

*"Still, a lot can be learned through such simulations."*

- Figures: the same colors are used in different figures with a different meaning. Maybe using different color palettes for different variables would improve the readability of the paper.

*Thanks for this suggestion. We are using the same colour palette as it is one that ensures good readability also to colour blind people and when for example printed in grey scale. Since some other palettes exist that also fulfil these requirements, we changed the colours in some figures, so that each palette represents a different variable.*

- Lines 293-297: I can't find evidence of the statement here in the figures. What figures are the Authors referring to?

*Thanks for this comment. We added this information (l. 310):*

*"For heavy-tailed rainfall distributions, a more variable runoff generation at the same level of mean storage capacity tends to lead to a lower flood peak shape parameter (Fig. 9, bottom-right). For light-tailed rainfall distributions, this is only true for medium catchment storage capacities while for larger storage capacities a more variable runoff generation leads to higher shape parameters (Fig. 9, bottom-left)."*

- Line 355: a space is missing.

*Added.*

**'Comment on hess-2024-181', Anonymous Referee #2**

In this manuscript, the authors present a study on the tail behaviour of peak discharges. Given the fact that observation data is too short and/or sparse  the approach of using a rainfall-runoff model of a synthetic catchment and time series from weather generator for such an in depth analysis is justified to overcome these limitations. The paper is well structured and written and the presentation of the results is of good quality. However, there are some points that should be addressed or clarified, especially with respect to the simulation model chain in section 2. More details and explanations about the weather generator and the rainfall-runoff model should be provided, e.g.:

*We thank the reviewer for the comments and thorough review. Below we address each of the points and state which changes we will make in the manuscript to address the comments.*

l. 132: What are these 678 locations? Points? Catchments? And what does the weather generator generate? Multi-site time series at points or distributed data? If it is point data, then how is the catchment precipitation derived? And how is the tempo-spatial variability of rainfall within a catchment accounted for?

*Thanks for pointing out that this is not clear enough yet. The 678 locations are points, but each of them is also taken as representative for a catchment. With the weather generator, multi-site time series at these points are generated. This point data is then considered to be representative for the respective catchment. In the rainfall-runoff model, the point data is therefore used as areal input, as described in l. 154. The 678 catchments are the smallest units we consider in our setup, so no further spatial variability within them is considered. To clarify, we changed this part as follows :*

*"Here, it is used to generate multi-site point data for 678 points in the synthetic catchment. Each of the points is considered to be representative for a sub-catchment and later taken as the areal input for the respective sub-catchment."*

l. 140 ff How is the "spatial dependence strength" of the E-OBS data set derived and which metrics are used for this? And is the spatial resolution of 0.1°/0.25° sufficient for this? Wouldn't it be better to derive this information from a radar data set?

*We use the E-OBS dataset version 25.0e (Cornes et al., 2018) to analyze spatial dependence in daily precipitation across Germany (Nguyen et al., 2024). Spatial dependence is quantified by calculating the pairwise cross-correlation of precipitation between grid cells, modelled as an exponential decay with distance, where the decay coefficient $k$ denotes dependence strength. For Germany, $k \approx 0.0025$ indicates medium-strength dependence. We categorize other strengths as follows: $k=0.01$ for weak, $k=0.005$ for medium-weak, $k=0.0015$ for medium-strong, and $k=0.001$ for strong dependence. The E-OBS dataset's resolution is adequate for capturing broad spatial patterns in precipitation dependence, balancing spatial detail with extensive temporal coverage. Although higher-resolution radar data could reveal finer-scale details, E-OBS is well-suited to our study due to its consistent quality and long-term availability across Europe. We changed the manuscript as follows (l.142):*

*"To generate P time series with different spatial variability, the spatial dependence strength of E-OBS data in Germany (Cornes et al., 2018) is derived. Spatial dependence is quantified by calculating the pairwise cross-correlation of precipitation between grid cells, modelled as an exponential decay with distance, where the decay coefficient k denotes dependence strength. For Germany, k is around 0.0025 and labelled as medium (M) dependence strength."*

l. 146 ff. Is the rainfall-runoff model lumped or distributed? If distributed, which spatial resolution does it have?

*As mentioned above, the 678 sub-catchments are the smallest units we consider. For each of them, the rainfall-runoff model is run in a lumped way. We added this information:*

*"The model is run in a lumped way on each of the 678 sub-catchments of the large synthetic catchment (101,588 km2)."*

I also wonder if a daily temporal resolution is meaningful for catchments smaller than ~1000 km². Most run-off generation processes in smaller scales are sub-daily. What implications does this have and how does this influence the results?

*Please see our response to the second point of the first reviewer. We added in the discussion (l.405):*

*"For some small catchments, running the model at an hourly instead of a daily resolution would have been more appropriate as concentration times might be sub-daily. However, this would have increased the computational costs drastically. Especially for the weather generator, generating time series of 7000 years for the given setups in an hourly resolution did not seem feasible. Further, a higher temporal resolution would have limited the data available for setting up and calibrating the weather generator. With the aim of balancing between accuracy and computational feasibility, the weather generator and rainfall-runoff model were run at a daily scale, while the data was disaggregated to a higher temporal resolution for the river routing."*

**Section 2.3**

l. 203 For fitting the distributions the L-moments are used. How do others methods like MLM influence this fitting, especially with respect to the tails?

*Please see our response to the first comment of the first reviewer. We added in the manuscript (l. 214):*

*"L-moment fitting has been evaluated to be reasonably efficient in parameter estimation compared to the maximum likelihood method (Hosking, 1990), while being a lot less computationally expensive."*

l. 209 How many sub-catchments on average are used for calculating the CV ?

*The number of sub-catchments per analysed catchment range from 3 to 200 with an average of 14. We added in l. 208:*

*"This results in 163 catchments (Fig. 2) which consist on average of 14 sub-catchments and for which all subsequent analyses are conducted."*

l. 210. How are rainy days defined?

*We added (l.226): "Rainy days are defined as all days on which it rained in at least one sub-catchment (P > 0 mm)."*

**Section 4**

l. 304 ff Could the difference between the shape parameters be due to the fact that one is based on real world observations and one on a synthetic study?

*Yes, this could very well be the case and goes along the line of argumentation that we present. We state that the difference could be related to the modelling setup (l. 306), which ultimately boils down to the difference between a modelling study and one based on real-world observations. In the real-world data, all relevant processes are embedded, while in the simulations we are analysing specific setups that might neglect certain processes. We changed the respective sentence as follows:*

*"This could be a regional difference, or related to differences between real-world data and simulations with a specific modelling setup: ..."*

**Minor comments:**

l. 19 "occurrence" is maybe not the right word in this context, "probability" would seem more appropriate as this does not imply a causality...

*We understand the point of the reviewer, but feel that saying "making the probability more likely" would not make sense, as the probability is a measure of how likely something is and can therefore not itself be more or less likely. To reduce the impression auf causality we rephrased as follows:*

*"The upper tail of a distribution is called heavy when it decreases slower than exponentially, indicating that the occurrence of extremes is more likely than for an exponentially receding tail."*

l. 335 Please explain (and cite) the REGNIE data.

*Since we do not use the data ourselves, but only cite a study that does, we keep this very brief and added:*

*"The range of rainfall variability in Wang et al. (2023) is based on the estimated spatial variability for 175 German catchments based on REGNIE data, a rainfall field interpolated from point observations (Rauthe et al., 2013)"*

***Additional references***

*Hosking, J. R. M. (1990), "L-Moments: Analysis and Estimation of Distributions Using Linear Combinations of Order Statistics," Journal of the Royal Statistical Society, Series B, 52, 105–124. https://onlinelibrary.wiley.com/doi/10.1111/j.2517-6161.1990.tb01775.x*

*Hosking, J. R. M., and Wallis, J. R. (1987), "Parameter and Quantile Estimation for the Generalized Pareto Distribution," Technometrics, 29, 339–349.*

*Nguyen, V.D., Vorogushyn, S., Nissen, K., Brunner, L., Merz, B.: A non-stationary climate-informed weather generator for assessing of future flood risks. Adv. Stat. Clim. Meteorol. Oceanogr., 2024 (accepted)*

*Rauthe, M., Steiner, H., Riediger, U., Mazurkiewicz, A., Gratzki, A., 2013. A Central European precipitation climatology - Part I: Generation and validation of a highresolution gridded daily data set (HYRAS). Meteorol. Z. 22 (3), 235–256. https://doi.org/10.1127/0941-2948/2013/0436*

*Richard M. Vogel, Simon Michael Papalexiou, Jonathan R. Lamontagne & Flannery C. Dolan (21 Oct 2024): When Heavy Tails Disrupt Statistical Inference, The American Statistician, DOI: 10.1080/00031305.2024.2402898*